# Acceleration Magnitude at Impact Following Loss of Balance Can Be Estimated Using Deep Learning Model

**DOI:** 10.3390/s20216126

**Published:** 2020-10-28

**Authors:** Tae Hyong Kim, Ahnryul Choi, Hyun Mu Heo, Hyunggun Kim, Joung Hwan Mun

**Affiliations:** 1Department of Biomechatronic Engineering, College of Biotechnology and Bioengineering, Sungkyunkwan University, Suwon 440-746, Korea; sanctified@skku.edu (T.H.K.); achoi@cku.ac.kr (A.C.); hhmoo91@skku.edu (H.M.H.); 2Department of Biomedical Engineering, College of Medical Convergence, Catholic Kwandong University, 24 Beomilro 579 Beongil, Gangneung, Gangwon 25601, Korea

**Keywords:** falling, pre-impact fall detection, peak impact acceleration magnitude, deep learning, data augmentation, wearable fall protective device

## Abstract

Pre-impact fall detection can detect a fall before a body segment hits the ground. When it is integrated with a protective system, it can directly prevent an injury due to hitting the ground. An impact acceleration peak magnitude is one of key measurement factors that can affect the severity of an injury. It can be used as a design parameter for wearable protective devices to prevent injuries. In our study, a novel method is proposed to predict an impact acceleration magnitude after loss of balance using a single inertial measurement unit (IMU) sensor and a sequential-based deep learning model. Twenty-four healthy participants participated in this study for fall experiments. Each participant worn a single IMU sensor on the waist to collect tri-axial accelerometer and angular velocity data. A deep learning method, bi-directional long short-term memory (LSTM) regression, is applied to predict a fall’s impact acceleration magnitude prior to fall impact (a fall in five directions). To improve prediction performance, a data augmentation technique with increment of dataset is applied. Our proposed model showed a mean absolute percentage error (MAPE) of 6.69 ± 0.33% with r value of 0.93 when all three different types of data augmentation techniques are applied. Additionally, there was a significant reduction of MAPE by 45.2% when the number of training datasets was increased by 4-fold. These results show that impact acceleration magnitude can be used as an activation parameter for fall prevention such as in a wearable airbag system by optimizing deployment process to minimize fall injury in real time.

## 1. Introduction

Balance posture controls the center of mass within the base of body support for equilibrium state by maintaining interactions among visual, vestibular, and somatosensory systems [1,2,3]. When such interactions are broken, the center of mass will rapidly go toward the ground, causing a fall during which human body segments can hit the floor [4]. According to the World Health Organization, 32% of elders aged over 70 years will experience falls each year [5]. It has been reported that 30% of elders aged over 65 years in the United States have at least one fall per year [6]. Falls are generally caused by decreased physical strength, posture control, and balance-keeping ability, leading to injuries such as hip joint injuries with serious pain and sequelae of long-term hospitalization [7]. Aging causes deterioration of lower extremity’s muscle strength, limited range of motion, and cognitive impairment in the case of elders [8]. Decreased balance ability may lead to falls [9]. Therefore, detecting falls with appropriate response has become an important social issue in an aging society [10].

To minimize injuries caused by falls, many studies on fall detection are being actively conducted. In general, fall detection is divided into pre-impact detection and post-fall mobility detection in previous studies [11]. Post-fall detection has the advantage of minimizing “long-lie”, which refers to staying on the floor for a long time. It can secure a rapid medical support after a fall. However, it is fundamentally impossible to prevent injuries caused by a fall with post-fall detection [12]. On the contrary, pre-impact fall detection can detect a fall in advance before a body segment hits the ground. It not only has the advantage of post-fall detection, but also can interlock the protective system to directly prevent injuries due to falls [13]. Although research on protective systems such as wearable airbag devices is at its early stage without any commercialized systems, it is very clear that injury prevention is possible by integrating pre-sensing devices [14]. Pre-impact fall detection is a more challenging research than post-fall mobility detection. Thus, interest in pre-impact fall detection research is increasing [15].

Fall can be divided into four phases: pre-fall, critical, post-fall, and recovery. The critical phase is a short process from the point when the center of gravity of the human body goes toward the ground to the point when it gets vertical impact due to hitting the floor [16]. At the last moment of the critical phase, the peak of upper body acceleration due to impact appears. It is regarded as a fall impact. Various fall experiments have suggested that the acceleration value at impact is approximately 4–11 m/s^2^ with different ranges depending on the type or direction of the fall [17]. It has been reported that impact acceleration of a backward fall is approximately 25% greater than that of a forward fall through kinematic analysis of human body segments [18]. Such impact acceleration peak value is one key measurement factor that can affect the risk of injury [19]. It can be used as a design parameter of wearable protective devices for injury prevention [20]. When operating an airbag system in a fall situation, the relative lower air pressure compared to impact magnitude causes a large deformation of foam and contact area with the floor, thereby allowing the body segment to directly contact the floor [21]. In contrast, there is a risk of injury when a relatively high pressure causes a secondary impact due to recoil between the foam and body segment [22]. To prevent possible injuries, it is necessary to optimally control the airbag expansion rate according to the impact magnitude for pre-impact fall detection. A study that estimates the peak impact acceleration is needed.

To predict falls, cameras and sensor technologies have been applied in previous studies [23]. A dedicated wearable inertial measurement unit (IMU) is generally used for active response to fall situations [24,25]. An IMU has a small size with fast data processing capability. It is a low-cost sensor module that uses low power with the advantage of interworking with other systems [26]. It has been reported that acceleration or angular velocity signals are suitable for distinguishing human body movements, making them very suitable hardware for pre-impact fall detection devices [11]. When using an IMU module, determining the number of sensors and attachment location is one important issue. Increasing the number of IMU modules attached to the body can be helpful for detecting various motions by acquiring signals of multiple body segments. However, there might be a problem of convenience. Most of previous studies have used very few IMU modules due to the problem of technical complexity when collecting data and time-series synchronization between modules [27]. In addition, it has been suggested that attaching an IMU sensor to the waist can gather the most consistent signals in terms of accuracy for detecting a fall among sensor signals from various locations [28]. Therefore, various attempts have been made to predict a fall in advance using one IMU signal attached to the waist. 

Algorithms for predicting falls prior to impact using a single IMU signal are largely divided into methods using thresholds or artificial intelligence (AI) to detect them [27]. One study has reported that the system can detect the signal as a fall when the acceleration of IMU worn on waist is less than 3 m/s^2^ and the angular velocity is greater than 0.52 rad/s, with an accuracy of fall-detection of approximately 93% [29]. However, the method of predicting falls using a threshold is inconsistent depending on types of fall situations, age, and so on. Thus, improving the accuracy of fall detection has a limitation [30]. To overcome such limitation, using machine learning models to increase the accuracy of pre-impact fall detection has been proposed based on various experimental data for training. Several studies have been carried out to detect falls before impact using various machine learning techniques such as support vector machine (SVM), hidden Markov model (HMM), and discriminant model (DM). Highly correlated parameters among raw signals of acceleration and angular velocity as input are selected. They show accuracies of approximately 93 to 97% [31,32]. In addition, several studies have applied deep learning technology to reduce the dependence of feature extraction from input signals, further improving the accuracy for fall detection. A multi-class fall classification model architecture that includes falls in various directions utilizing three kinds of deep learning algorithms has been developed, showing a pre-impact fall detection accuracy of 99% [33]. convolutional neural network (CNN) and long short-term memory (LSTM) deep learning techniques have also been utilized to develop a pre-impact fall estimation model with accuracy up to 98.7% [13]. 

To the best of our knowledge, most studies using an IMU device to estimate falls prior to impact have focused on accuracy improvement. No attempt has been made to predict impact acceleration peak magnitude due to body impact. As mentioned above, an impact peak magnitude is used as a wearable fall protective device design parameter that is essential for preventing an injury. Therefore, the purpose of this study was to propose a deep learning architecture that could predict an impact acceleration magnitude in the event of a fall using one IMU sensor signal worn on the waist. Another objective of this study was to verify the performance of the proposed learning model using experimental data of falls in various directions.

## 2. Materials and Methods

### 2.1. Application to Multi-Class Pre-Impact Fall Impact Accelerometer Magnitude Prediction Model

As described above, there are four distinct phases of falls (Figure 1). The pre-fall phase is when normal activities occur. The critical phase is when fall event happens. From acceleration and angular velocity data collected from experiments, the start of activity is referred to as S within the critical phase, the impact of fall is called I, and the end of fall is referred to as E. The post-fall phase is when a subject has no movement, lying down on a floor. Lastly, the recovery phase is when a subject stands up and shows movement [34,35]. A fall monitoring system can be divided into two main systems depending on the detecting point during critical phase of fall. In this research, we focus on the detection of a fall impact acceleration magnitude prior to impact point within the critical phase. For a fall prevention system such as a wearable airbag system to be optimally activated, pre-impact fall impact peak acceleration magnitude should to be accurately predicted using different types of fall [33].

### 2.2. Data Collection

#### 2.2.1. Apparatus

An IMU containing a data collecting device was developed to collect tri-axial accelerometer and angular velocity signals for pre-impact fall impact acceleration magnitude prediction. The size of the device was 37 mm (width) × 60 mm (length) × 17.5 mm (height) (Figure 2a). An mpu6050 (InvenSense Inc., San Jose, CA, USA) that could collect tri-axial accelerometer and angular velocity data was used. It contained a 16-bit analog-to-digital converter for digitizing accelerometer and gyroscope outputs. The range of data output was ±16 g for the accelerometer and ±2000 deg/s for the gyroscope. The number of sample rates for data collection was chosen at 40 Hz. Wireless connection between the data collecting device and the workstation was done using a Bluetooth V4.0 BLE module (HM-11, JN Huamao Technology Co., Jinan, China). The model of microcontroller (MCU) was an STM32F103CB (STMicroelectronics, Geneva, Switzerland). The power supply source was a lithium polymer battery (3.7 V, 720 mAh) as shown in Figure 2b [36]. The circuit diagram of the developed IMU device for collecting raw signals is shown in Figure 2c. The device is connected to the workstation through the Bluetooth module. A data collecting software, C# based windform, was utilized to collect data from the workstation.

#### 2.2.2. Subjects

Twenty-four healthy participants (14 males and 10 females, age: 22 to 34 years; height 1.57 to 1.83 m; weight: 46 to 81 kg) without any musculoskeletal disorders volunteered to participate in this study. All volunteers were students at Sungkyunkwan University. They were recruited through advertisements. All participants provided informed written consent. All experiments were performed in accordance with relevant guidelines of Sungkyunkwan University.

#### 2.2.3. Experimental Protocols

All fall experiments were performed under the direct supervision of support staff. The experimental protocol was prepared to mimic realistic falls often occurring among elders based on previous studies [16,23,33]. In this study, ten different fall types (forward fall, backward fall, leftward fall, rightward fall, fall on ground, fall on ground while rising, forward fall while walking, backward fall while walking, tripping, and slipping) were collected. A total of 1278 datasets (number of trials: 113–140) for each experiment x number of activities (*n* = 10) were collected. Fall data were sorted for five different directions of falls depending on the direction of waist. For instance, a tripping fall type meant that when the device’s direction of z axis of accelerometer signal was facing toward on the ground. This type of fall was grouped into front direction falls.

#### 2.2.4. Data Collections and Processing

Pre-impact fall impact accelerometer magnitude prediction at five directions was started by collecting data from wearable IMU sensors. Recruited subjects were asked to wear the developed IMU sensor on the left side of the pelvis. Time-series tri-axial accelerometer and gyroscope data (a total of six channels) were collected for each experimental dataset. One of the most important processes for a classification or a prediction model is to label data collection. In this research, we manually labeled the frame for the impact of fall depending on the direction of fall and the impact acceleration magnitude of the accelerometer.

### 2.3. Feature Extraion from Raw Inertial Measurement Unit Signal

In the flow of human activity recognition, feature extraction is the most important phase. It improves the performance of a system by extracting feature vector that can discriminate activities. For continuous data such as sensor data, feature extraction or selection is a very challenging task [37]. In this research, we calculated four additional features from raw tri-axial accelerometer and angular velocity data related to translational or vertical magnitude value based on previous studies as shown in Table 1 below. Input data matrices size was the window size by ten features. One of these features, sum of vector magnitude, was the square root of sum of tri-axial accelerometer which was the impact accelerometer magnitude value to be predicted using our proposed model.

### 2.4. Data Augmentation

Data augmentation techniques are widely utilized in computer vision to introduce new data samples between pairs of training datasets. Classification with time-series data with a small number of samples may lead to an overfitting problem [38]. To overcome this limitation, time-series data augmentation technique is applied in this study. It allows us to generate synthetic training datasets to train the deep learning network with a large number of training datasets for better performance. There are several techniques for augmenting data such as stretching, shrinking, and removing some data points [39]. Selecting which data augmentation technique depends on the application. For instance, some applications such as vital time-series data of patients can remove important information which will decrease classification accuracy. In this study, we applied IMU sensor attached to the left iliac crest by fixing the device on the belt which cannot be rotated. Therefore, the following three data augmentation techniques were used: (i) jittering by adding mechanical noise to increase the robustness of training model with white gaussian noise applied to the training dataset; (ii) scaling by changing the magnitude of data by multiply random scale value; and (iii) time-warping by changing time intervals of data, thus changing temporal characteristics of sensor data by shortening or stretching with a random warping ratio [40]. In real-life scenario, a person’s active daily living or falling activity can be performed with various speeds or raw signals of the sensor are collected with noise. For instance, a person can walk slower or faster than a normal gait speed. This variability of a subject’s movement needs to be considered to improve robustness of our trained model. Therefore, one of data augmentation techniques, time-warping, is considered and additional training dataset is generated by multiplying pre-determined warping value to lengthen or shorten the original sensor signal. Based on the previous research, each augmentation technique was applied to generate a 4-fold increase of training data using four different levels or values of each technique [41]. 

### 2.5. Deep Learning Network

#### 2.5.1. A Bi-Directional Long Short-Term Memory

Activity recognition of human movement which is a complex motor movement with a high variance is done using a classical time series classification method. Previous studies have used deep learning methods such as CNN and recurrent neural network (RNN). RNN was developed to classify sequential time series data [42]. Temporal layer of RNN holds sequential information. It learns using hidden units of recurrent cells. It gets updated and computes current hidden state by estimating the next hidden state. To overcome the limitation of RNN, a LSTM model is developed to hold and capture activity sequences from gates and memory cells in this study. A LSTM cell is composed of an input gate it, a forget gate f_t_, a cell c_t_, and an output gate h_t_ as defined in the formula below.

i_t_ = σ (W_xi_^xt^ + W_hi_^ct-1^ + b_i_)
(1)

f_t_ = σ(W_xf_^xt^ + W_hf_^ht-1^ + W_cf_^ct-1^ + b_f_)
(2)

f_t_ = f_t_⊗c_t-1_ + i_t_ ⊗ tanh(W_xc_^xt^ + W_hc_^ht-1^ + b_c_)
(3)

o_t_ = σ(W_xo_^xt^ + Wh_o_^h^_t-1_ + Wc_o_^ct^ + b_o_)
(4)

h_t_ = o_t_ ⊗ tanh(c_t_)
(5)
where ⊗, σ(x), W_αβ_, and b_β_ were product, sigmoid function, weight matrix between α and β, and bias of β with β ∈ {i,f,c,o}, respectively [43]. A deep learning LSTM model allows different sizes of input vector for training and testing processes, unlike other classification algorithms such as a neural network. 

#### 2.5.2. The Flow of the Impact Fall Prediction Model Using Deep Learning

After the feature augmentation process to increase training datasets, newly generated datasets were separated by a window size of 20 frames equivalent to 0.5 s using a moving window technique. Ten features (i.e., sum vector magnitude of tri-axial accelerometer) were calculated as mentioned above in each window. Therefore, the dimension of each dataset was set as 10 ∗ length of each window (i.e., 10 features × 20 frames for no-augmentation technique applied dataset). These datasets and corresponding labeled data were used as inputs and outputs for training and testing LSTM networks. To train a deep bidirectional LSTM model, collected datasets were divided into training and testing datasets at a ratio of 1:1. For example, there were a total of 1278 raw datasets. These datasets were randomly divided so that 639 datasets were used for augmentation and training while the remaining 639 datasets were used for testing our proposed model.

The overall architecture of our proposed model for predicting impact acceleration magnitude during a fall is shown in Figure 3a. The main architecture is composed of a sequence input layer, two bi-directional LSTM layers, a dropout layer, a fully connected layer, a softmax layer, and a regression layer. Two bi-directional LSTM layers were added between the sequence input layer and the fully connected layer. The one dropout layer was added between two bi-directional LSTM layers. It was added to decrease overfitting of the bi-directional LSTM layer. Predicting fall impact acceleration magnitude is performed for the regression layer. Detailed architecture of the bi-LSTM network is shown in Figure 3b.

Previous studies have reported that optimal hyper-parameters of a deep learning network need to be determined. This will increase the performance of training model while reducing the overfitting problem [44]. In this study, ten different types of hyper-parameters and their values were chosen. For instance, when the initial learning rate is too low, then the training time is too high. If the learning rate is too high, then training might reach a suboptimal result. Additionally, we added a drop out layer to the fall impact acceleration magnitude prediction model. The dropout layer randomly sets input elements to zero and then scales remaining elements. This operation effectively changes the network architecture between training iteration and helps prevent the network from overfitting. If the dropout rate is high, then more elements are dropped during training [45,46]. Specific hyperparameters for the bi-directional LSTM network are shown in Table 2 below.

In this study, we utilized MATLAB program (R2019B, Mathworks, Natick, MA, USA). Its deep learning toolbox is utilized for developing and training the deep-learning network to predict impact acceleration magnitude during a fall.

### 2.6. Performance Measures

In this research, the mean absolute percentage error (MAPE) was adopted to determine the performance of the multi-direction pre-impact fall impact peak of the acceleration magnitude prediction model. MAPE was used to measure the relative error between the observed value and the predicted value to reflect the specificity of an average predicted value [45,47]. It was calculated as follows:(6)$$\mathrm{Mean}\;\mathrm{absolute}\;\mathrm{percentage}\;\mathrm{error}=\frac{1}{n}\sum_{j=1}^{n}\left|Y_{j}-{\hat{Y}}_{j}\right|\times 100\%$$
where *Y_j_* denoted the observed impact accelerometer magnitude and ${\hat{Y}}_{j}$ denoted the predicted impact accelerometer magnitude. Results were analyzed using regression analysis to estimate relationships between observed and predicted values. The slope was compared against a 1:1 line. R was calculated from the linear model. Additionally, obtained results were compared and analyzed through ANOVA with Turkey’s post-hoc test to determine differences between groups. The significant level was set at *p*-value < 0.05. All statistical analyses were conducted using PASW Statistics 18 (SPPS Inc., Chicago, IL, USA). Results were analyzed using MATLAB program (R2019b, MathWorks, USA).

## 3. Results

The developed IMU and data collection firmware allows a sensor attached to the waist to send tri-axial accelerometer and angular velocity signals to the workstation. The direction of the axis of acceleration was different for each fall type. For instance, the z-axis of a front fall showed a positive value whereas a back fall with z-axis magnitude had a negative value. Received raw sensor signals were used to calculate features such as resultant impact acceleration magnitude which was calculated by square root of three axes of accelerometer signals as shown in Figure 4.

Generation of new training datasets based on raw signals with application of each augmentation approach is shown in Figure 5. Three data augmentation techniques ((i): jittering; (ii): scaling; and (iii): warping) were applied to raw data at four different levels. Increasing the jittering value will create a new signal to become noisier than the raw signal as shown in Figure 5b. Additionally, as the scaling value increases, the magnitude of IMU signal gets bigger, meaning that the impact acceleration magnitude value is increased as shown in Figure 5c.

Figure 6 presents results of a fall impact acceleration magnitude prediction model proposed in this study. A bi-directional LSTM algorithm for deep learning was applied to predict the magnitude of acceleration at impact point since collected accelerometer and angular velocity signals were of time-series type. Time-series data near a point when a subject lost balance shown in red were used as input data for the LSTM model to predict the observed value shown in green dot which was the actual impact acceleration magnitude value. In this study, five different directions of fall (front, back, left, right, and straight) experimental data were collected. Additionally, we generated new dataset using data augmentation technique at four different values to increase datasets by 4-fold. As shown in Figure 6, the impact acceleration magnitude value varied throughout datasets since characteristics of person falling, direction of falling, and anthropometric measurements were different. For instance, heights and weights of participants of this study were different. The activity prior to falling such as standing or walking showed different acceleration values which could affect the falling speed. The time between the start of the critical phase to impact was also different. Predicted results without augmentation approaches from our proposed model for different levels of acceleration magnitude were determined. MAPE values for all augmentation approach datasets with a 4-fold data size increment in trials 1, 2, and 3 were 6.9%, 7.5%, and 7.8%, respectively; lower than those without an augmentation approach (27.8%, 29.1%, and 30.5%, respectively).

Additionally, we analyzed how well our proposed model could predict the impact acceleration magnitude using a linear regression model for five fall directions. Results are shown in Figure 7. Our proposed model results showed r values of 0.7, 0.76, 0.75, 0.75, and 0.814 for front, back, left, right, and straight falls, respectively, when all-data augmentation approach was applied with onefold increment dataset. Compared to onefold dataset results, the fourfold dataset increment model was able to predict impact acceleration magnitude significantly better, with r values of 0.94, 0.95, 0.92, 0.92, and 0.95 for front fall, back fall, left fall, right fall, and straight fall, respectively.

As shown in Figure 8, the all-data augmentation technique dataset achieved a significant (*p* = 0.01) decrease of average MAPE by 55.8% compared to the no-data augmentation dataset. Similarly, there was a significant (*p* = 0.01) difference in average MAPE value by 32.4% between results of single augmentation approach-applied dataset (jittering, scaling, and warping) and all-data augmentation technique-applied datasets. However, there was no significant difference in the performance between each single-data augmentation group and the all-data augmentation group. In summary, data augmentation technique seems to be effective for predicting impact acceleration magnitude prior to a subject’s impact on the ground.

Figure 9 presents average MAPE values of fall impact acceleration magnitude predicted by our proposed model with increasing number of training datasets. In this study, we collected 1275 fall datasets for five different types of falls and increased the number of datasets up to 4-fold by applying different feature augmentation techniques. From post hoc analysis, the group with fourfold and all-data augmentation technique applied dataset showed the lowest average MAPE value of 7.4% which was 45.2% lower than the no-fold dataset group with a *p* value of 0.001. There was no significant difference in MAPE value between the no-fold group and the onefold or the twofold group. The average MAPE and standard deviation of MAPE when all data augmentation technique is applied also decreased significantly, meaning that the prediction model could predict impact acceleration magnitude with less variance for a more stabilized system. The overall performance of our proposed model is shown in Table 3 below.

## 4. Discussion

Fall peak impact acceleration magnitude is an important factor for determining the risk of fractures such as head injury criterion (HIC) value and a parameter for designing and developing fall prevention devices such as a wearable inflate airbag system [48]. Previous studies have reported that an impact magnitude is affected by many uncertainties such as fall direction, impact site, and pre-impact movement prior to fall, leading to a wide range of impact acceleration values [49]. These uncertainties require us to create biomechanical models or optimization processes when designing an airbag system. They might decrease the performance of predicting an impact magnitude. However, our model allows the prediction of impact acceleration magnitude by using a deep learning model with a single IMU raw sensor data, showing a high performance. Additionally, when our proposed model was implemented to fall prevention devices such as a wearable inflation airbag system along with a classification model, it could classify human activity by normal vs. fall with different directions. This will lead to the development of a full intelligent fall protective system by detecting fall itself and fall severity to reduce fall injuries.

Recently, intelligent airbag systems are developed and applied in the field of automobile. An airbag sensing system can adjust the pressure of an airbag called a low-risk deployment to protect passengers from pressure dispersion of an airbag [48]. The airbag system composed of mechanical release system allows compressed CO_2_ release by lock release signal from an MCU. The CO_2_ gas is then transmitted to inflate the airbag. Based on compressible-fluid mass flow rate equations, mass flow rate, area of airbag, pressure of airbag, and inflation time need to be accurately calculated when designing a wearable airbag system [49]. For instance, an expandable airbag helmet shows a high risk of bottom-out as the impact acceleration increases if the pressure of airbag is low inside. This will ultimately increase HIC values. Therefore, material, thickness, size, and pressure of airbag should be optimized and controlled accurately in real time depending on the impact magnitude to reduce the risk of injury for a wearable airbag [21].

Fall impact acceleration showed that the magnitude prediction model prior to the actual impact of the subject had an average MAPE of 6.69 ± 0.33% with an average r value of 0.93 when all-data augmentation technique was applied with a 4-fold data size increment (Table 3). The meaning of our results cannot be fully evaluated since it is the first study to develop a model that can predict impact acceleration magnitude using deep learning and wearable sensor signals to the best of our knowledge. For instance, an actual impact acceleration magnitude of 5.0 m/s^2^ with an average MAPE of approximately 6.7% will give a predicted acceleration magnitude ranging from 4.7 to 5.3 m/s^2^. In a real-life situation where impact acceleration magnitude ranges from 3 to 10 m/s^2^, the severity of a fall impact can be divided into approximately ten classes considering the predicted error from our results. This grading value can be meaningful for applying designed control parameters for fall protective devices in the future based on our study.

Previous research has reported that the number of training datasets needs to be large with heterogeneity characteristics. To do so, time-series data augmentation is a new emerging research area to increase high quality training dataset. The data augmentation technique has an advantage of simulating a real-world training dataset by transforming raw data. It can reduce labor-work for gathering a large number of datasets of falls [38]. Additionally, the synthetic training data by applying data augmentation technique can minimize over-fitting effects, increase generality of unseen data and remove bias from the trained model due to imbalance volume of training data. There are several other techniques such as rotation for data augmentation. In this research, we applied three data augmentation techniques (scaling, time-warping, and jittering). However, our research used an IMU sensor fixed on the waist with the switch heading forward. We did not apply the rotation data augmentation technique. Previous research has mentioned that increasing the number of datasets using data augmentation technique can boost performance, increase generalization to unseen data, and reduce the chance of overfitting [50]. Our results showed that data augmentation technique could improve the performance of fall impact acceleration magnitude prediction than no data augmentation. There was no significant difference in MAPE value between single data augmentation groups. However, the pattern of our results showing MAPE decreased with increasing number of datasets and application of data augmentation technique was similar to previous studies [51]. When the number of datasets was significantly increased, the average MAPE for all five directions of falls decreased significantly. The standard deviation of MAPE also decreased, meaning that the prediction model could predict impact acceleration magnitude with less variance for a more stabilized system [52]. This can be interpreted that using deep learning with wearable IMU signal and data augmentation technique can increase the performance of a prediction model. 

Most of lab-based research have several limitations for practical application of their results. First, collected data used for training datasets were gathered from young healthy subjects. Fall activity was simulated on a soft pad surface. Thus, there are differences between laboratory situations and real-life situations. However, falling experiments with elders or real-life fall experiments could harm a subject, making them unsuitable [53]. Nonetheless, a real-life fall is more extreme than that in a laboratory simulated situation, although the developed model can perform well in a real-life situation [54]. Second, our study used time-series (sequential) signal data with a deep bi-directional LSTM network model to predict the impact acceleration magnitude. There are various deep learning methods such as autoencoder and CNN [37]. However, our study proposed a novel architecture that could predict an impact acceleration magnitude prior to fall. Within our proposed architecture, other deep learning algorithms could process sequential data. They can be applied in the future. A different deep learning algorithm or a combination of several deep learning architectures with optimization should be performed to compare the performance and find optimal parameters to minimize training errors.

## 5. Conclusions

In conclusion, an impact acceleration magnitude prediction model prior to impact on the ground using a single IMU sensor-based deep learning is proposed in this study. A bi-directional LSTM algorithm allows impact magnitude prediction for five different types of fall with application of data augmentation techniques along with increasing number of datasets. We found that applying augmentation techniques with increased number of training datasets for a deep bi-directional LSTM network showed an average MAPE of 6.69 ± 0.35%. These results can be used to develop a fall prevention intervention such as a wearable hip airbag system by optimizing the pressure or air flow rate for deploying an airbag to minimize injury caused by a fall in real time and contribute to the design of rehabilitation programs.

## Figures and Tables

**Figure 1 sensors-20-06126-f001:**
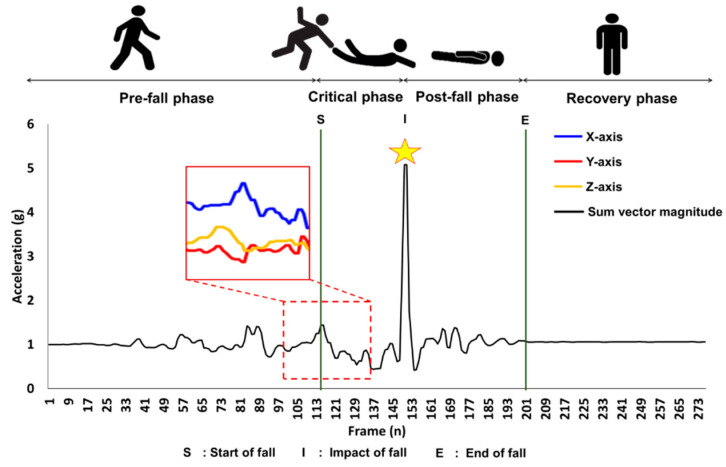
Description of four phases of falls. A subject loses balance during the critical phase of fall and the body heads down to ground which shows a direct impact during a fall. The peak of sum vector magnitude at the impact during critical phase of fall is indicated by a star.

**Figure 2 sensors-20-06126-f002:**
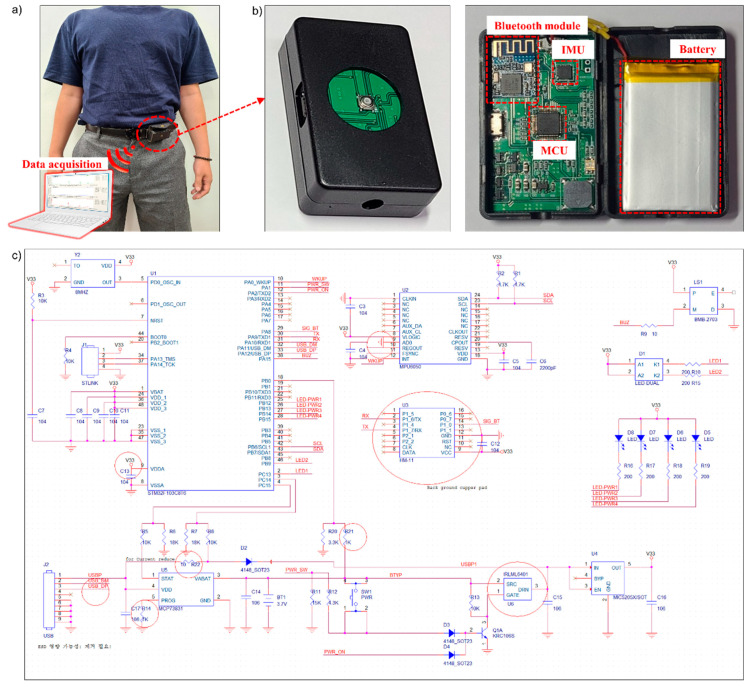
(**a**) A custom designed inertial measurement unit (IMU) sensor placed on the left anterior iliac crest of the pelvis of a subject; (**b**) Raw data are sent and received from the Bluetooth module and the receiver of a workstation; (**c**) Diagram showing the custom designed IMU circuit diagram.

**Figure 3 sensors-20-06126-f003:**
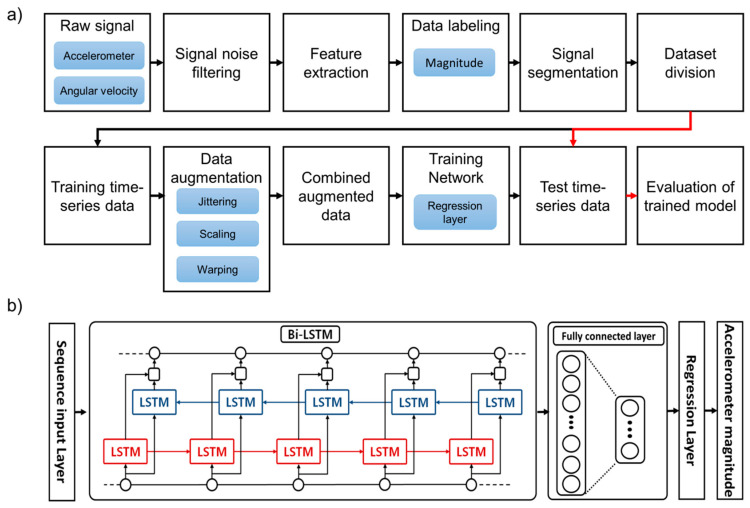
(**a**) Overall architecture for predicting impact acceleration magnitude during fall; (**b**) Detail architecture of long short-term memory network for fall accelerometer magnitude prediction.

**Figure 4 sensors-20-06126-f004:**
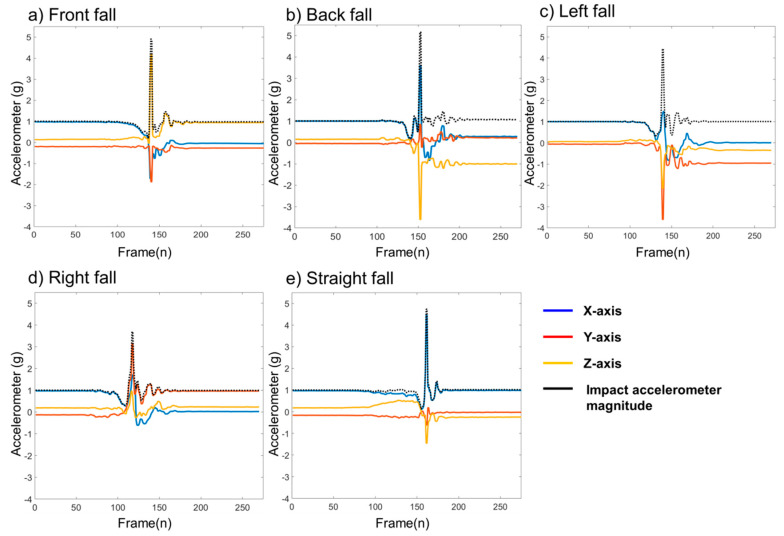
Tri-axial accelerometer collected from an inertial measurement unit (IMU) for five types of falls. (**a**) the IMU signal for forward fall; (**b**) the IMU signal for backward fall; (**c**) the IMU signal for leftward fall; (**d**) the IMU signal for rightward fall; (**e**) the IMU signal for straight fall.

**Figure 5 sensors-20-06126-f005:**
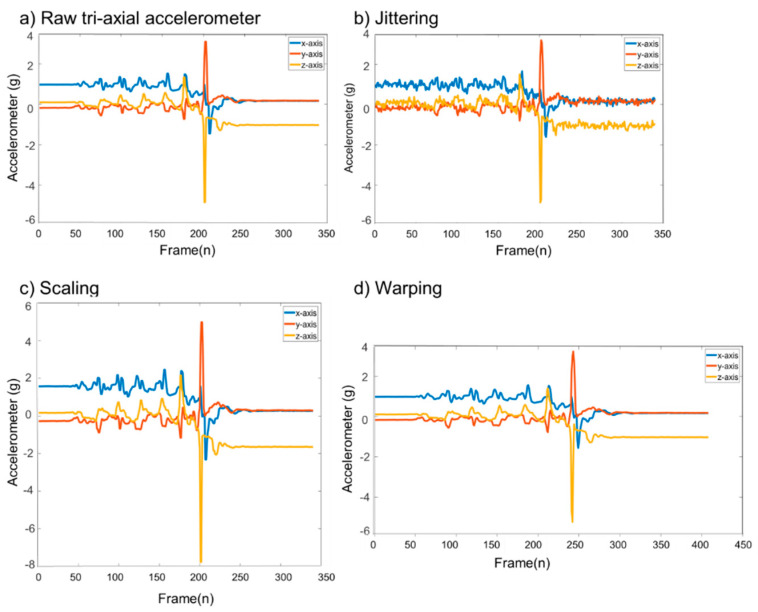
Three different data augmentation techniques applied to raw tri-axis accelerometer data (**a**): jittering data augmentation technique; (**b**): scaling data augmentation technique; (**c**): time-warping data augmentation technique.

**Figure 6 sensors-20-06126-f006:**
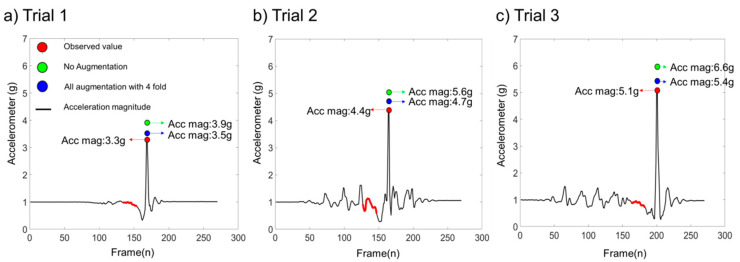
Pre-impact fall impact acceleration magnitude prediction at various levels of impact magnitude. (**a**) the impact acceleration magnitude prediction result for low level of impact; (**b**) the impact acceleration magnitude prediction result for mid-level of impact; (**c**) the impact acceleration magnitude prediction result for high level of impact.

**Figure 7 sensors-20-06126-f007:**
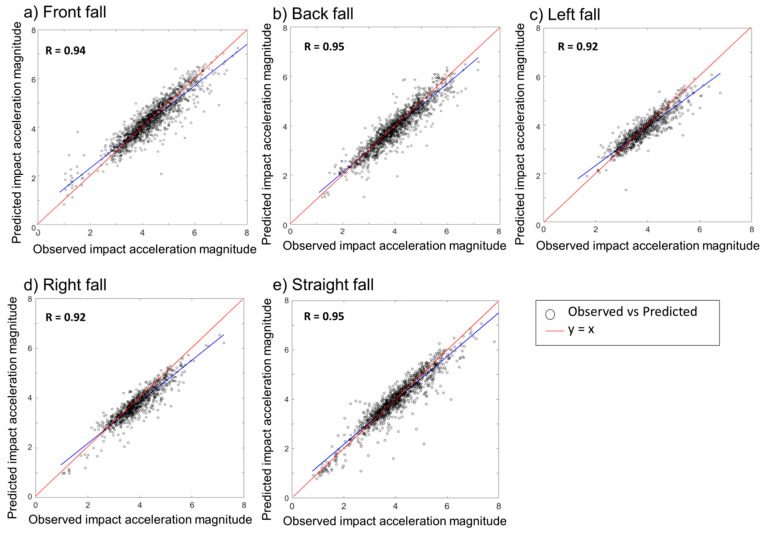
Regression analysis of pre-impact fall impact acceleration magnitude prediction at different directions. (**a**) the regression analysis of impact acceleration prediction for forward fall; (**b**) the regression analysis of impact acceleration prediction for backward fall; (**c**) the regression analysis of impact acceleration prediction for leftward fall; (**d**) the regression analysis of impact acceleration prediction for rightward fall; (**e**) the regression analysis of impact acceleration prediction for straight fall.

**Figure 8 sensors-20-06126-f008:**
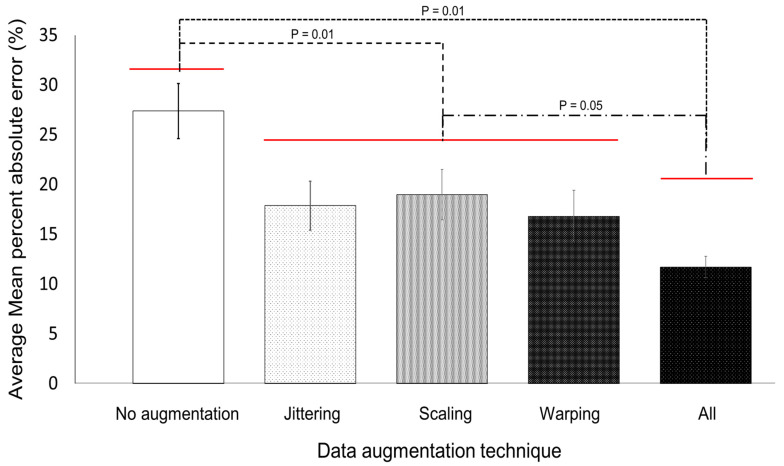
Average mean absolute percent error of fall impact acceleration magnitude prediction for three different data-augmentation techniques.

**Figure 9 sensors-20-06126-f009:**
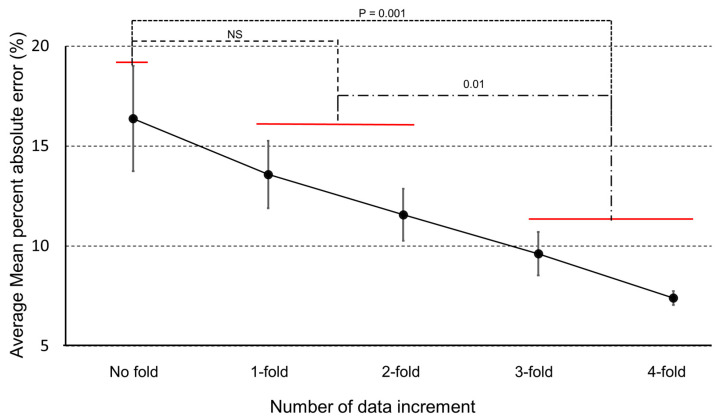
Average mean absolute percent error of fall impact acceleration magnitude prediction depending on the number of datasets with increasing fold of datasets.

**Table 1 sensors-20-06126-t001:** Types of features were extracted from inertial measurement unit sensor signals.

Number	Feature Types
1.	x-axis of raw accelerometer signal
2.	y-axis of raw accelerometer signal
3.	z-axis of raw accelerometer signal
4.	Sum vector magnitude of accelerometer
5.	Sum vector on horizontal plane of accelerometer
6.	Root mean square of sum vector magnitude
7.	x-axis of raw angular velocity signal
8.	y-axis of raw angular velocity signal
9.	z-axis of raw angular velocity signal
10.	Resultant angular velocity

**Table 2 sensors-20-06126-t002:** Bi-directional long short-term memory (LSTM) network model parameters.

Bi-Directional Long Short-Term Memory Network Architecture Training Option		
Number	Type of Parameters	Range of Parameters
1.	Number of hidden units	[100, 50]
2.	Maximum epochs	125
3.	Mini-batch size	64
4.	Weight initializer function	Glorot
5.	Solver	Adam
6.	Dropout rate	0.2
7.	Initial learning rate	0.01
8.	Gradient threshold	2
9.	Gradient threshold method	Global-l2norm
10.	L2Regularization	1 × 10^−5^

**Table 3 sensors-20-06126-t003:** Overall performance of our proposed fall impact acceleration prediction model (MAPE: mean absolute percent error).

		Front Fall		Back Fall		Left Fall		Right Fall		Straight Fall	
		MAPE	r	MAPE	r	MAPE	r	MAPE	r	MAPE	r
Raw	No-fold	27.6	0.2	34.5	0.12	33.5	0.22	30.1	0.28	17.3	0.32
	1-fold	33.6	0.25	22.5	0.29	20.8	0.46	23.3	0.14	15.3	0.52
	2-fold	29.6	0.28	18.2	0.33	16.6	0.33	17.3	0.21	14.7	0.51
	3-fold	25.5	0.30	14.5	0.25	19.1	0.1	13.7	0.1	10.7	0.5
	4-fold	22.8	0.35	12.5	0.55	12.4	0.32	10.7	0.24	6.9	0.60
Jittering	No-fold	36.9	0.16	32.1	0.18	39.1	0.16	33.1	0.18	16.9	0.32
	1-fold	30.2	0.40	19.2	0.50	21.6	0.38	21.6	0.57	15.3	0.59
	2-fold	30.3	0.50	16	0.65	19.9	0.57	16.9	0.44	13.1	0.62
	3-fold	29.1	0.70	13.8	0.76	17.1	0.78	16.2	0.67	9.99	0.83
	4-fold	26.1	0.84	12.2	0.81	11.1	0.88	11.3	0.85	7.56	0.92
Scaling	No-fold	34.9	0.11	31.8	0.17	23.9	0.20	19	0.18	15.6	0.37
	1-fold	34.9	0.53	20.1	0.69	15.7	0.55	15.3	0.64	12.8	0.79
	2-fold	25	0.57	13.4	0.65	16.2	0.69	15.6	0.64	12.4	0.58
	3-fold	25.5	0.67	10.9	0.71	13.6	0.75	8.9	0.67	7.9	0.76
	4-fold	25.3	0.87	7.2	0.87	9.7	0.81	8.3	0.69	7.1	0.89
Warping	No-fold	24.1	0.38	22.8	0.18	23.9	0.34	27.5	0.22	11.9	0.34
	1-fold	22.6	0.53	14.6	0.64	20.5	0.59	12.7	0.60	13.9	0.58
	2-fold	25.3	0.52	13.1	0.60	13.8	0.55	11.9	0.71	11.7	0.71
	3-fold	22.4	0.56	9.7	0.66	13.1	0.78	9.4	0.71	8.6	0.79
	4-fold	17.2	0.85	8	0.88	9.1	0.84	8.6	0.90	7.9	0.93
All	No-fold	17.4	0.29	16.9	0.33	15.6	0.37	13.1	0.19	25.53	0.26
	1-fold	14.4	0.70	15.8	0.76	11.2	0.75	12.1	0.75	13.6	0.81
	2-fold	12.0	0.78	12.8	0.80	9.1	0.78	11.8	0.72	12.5	0.86
	3-fold	11.2	0.82	10.3	0.86	8.6	0.81	8.3	0.86	10.9	0.89
	4-fold	6.9	0.94	7.5	0.95	7.2	0.92	7.9	0.92	7.3	0.95

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
