# Peer review of "Acceleration Magnitude at Impact Following Loss of Balance Can Be Estimated Using Deep Learning Model"

_sensors, 2020, doi:10.3390/s20216126_

Round 1

Reviewer 1 Report

The paper proposes a method to predict the impact acceleration magnitude during the pre-impact stage of a fall. The method is based on a sequential-based deep learning model and requires just one a single Inertial measurement unit sensor located at the waist.

The research is interesting and seems adequatelly conducted. However, the paper is clumsily written and should be carefully revised by their authors before publications:

  • There are some paragraphs that are pretty obscure (or even apparently "out of place") followed by an "editor's note" that has not been deleted. For example lines 208-209, lines 265-268, lines 287-289, and so on.
  • A lot of details (sometimes apparently irrelevant) are provided about some aspects (for example, "Networks were trained and tested using a window system with Intel(R) Core(TM) i7-5930K @ 3.50GHz, NVIDIA GeForce GTX 1080, and 8GB graphic card", is this really relevant?), whereas other key parameters or issues are not even mentioned (for example, the sampling rate of the data acquired by the inertial sensors or the tool/framework that has been used to develop and implement the deep-learning model).
  • The selection of parameters in table 3 is not properly explained. Are they heuristically setup? Are they (or some of them) default of typical values of LTSM model? is  there any criteria to choose these parameters (all or some of them)?
  • Some acronymn such as MAPE are changed along the paper (it is refered as MPAE at the end of the paper, or either MPAE is an acronynm you have not defined, but I believe it is the same than MAPE). Please be more careful with this.
  • For me, the process of data "time-warping" is not described well enough nor the cause of its benefits in the overall performance satisfactorily explained.
  • In general, the paper doesn't seem completely well-organized and some things seems out of place, some missing and some redundant and repeated over and over. I recommend the authors to re-read and rewrite sections 2.4, 2.5, 3 and 4  to improve the general organization of the paper. 

The last comment is just a suggestion: Please consider the release of your experimental dataset so that it is publicly available to other researchers.This is not only a fair pratice but also a good contribution to the research community on this subject, and quite important in order to further compare with new proposals in a future. of course, it's just a suggestion.

Author Response

We greatly appreciate the reviewer for his/her valuable time and constructive comments to improve the quality of our manuscript. Below please see the attachment for the response to the reviewer’s comments and suggestions.

Reviewer 2 Report

This manuscript uses a single Inertial measurement unit sensor on the waist (to collect tri-axial accelerometer and angular velocity data) and a sequential-based deep learning model to predict an impact acceleration magnitude after the loss of balance.  With the test on twenty-four healthy participants, five types of direction fall have been predicted with high accuracy.  When combined with fall prevention intervention such as a wearable airbag system, fall injury can be minimized.

1. Please define the abbreviations before their first occurrences in the abstract, main text, and figure captions.

2. What does SMV in Fig. 1 stand for?  Is it related to the magnitude of the acceleration?

3. The font in some figure plots (e.g., Fig. 2c) is too small to be legible.

4. Table 1 does not seem to be needed as the information can be directly given in the main text.

5. Please justify the choice of the parameters used in Table 3.

6. Please elaborate on the discussion of the data augmentation techniques presented in Fig. 5 and the corresponding results.

7. Please provide insights into the experimental results obtained in this study to reveal the underlying physics (e.g., why the augmentation with 4 fold shows a better prediction).

8. It is helpful to benchmark the performance of the results from this study against those in the literature reports to directly demonstrate the novelty.

Author Response

(The authors gave the same response as above.)

Round 2

Reviewer 1 Report

The authors have (mostly) addressed my comments and I think the paper is now suitable for publication. 

The release of experimental data was only a suggestion.